# Orthostatic hypotension in stroke/TIA patients: Association with new events and the effect of the NAILED intervention

**Joachim Ögren** *, **Thomas Mooe, Anna-Lotta Irewall**

Department of Public Health and Clinical Medicine, Umeå University, Östersund, Sweden

* Joachim.ogren@umu.se

## Abstract

### Background

Fear of orthostatic hypotension (OH) and a reported association with an increased risk of cardiovascular (CV) events may limit antihypertensive treatment after stroke/TIA. In the NAILED trial, systematic titration of antihypertensive treatment resulted in lower blood pressure (BP) and reduced the incidence of stroke. Our aim was to assess the association between OH and CV events or death in a stroke/TIA population and the association between group allocation in the NAILED trial and risk of OH during follow-up.

### Methods and findings

This post-hoc analysis included all patients with complete BP measurement at baseline in the NAILED trial (n = 814). OH was defined as a drop in systolic BP $\geq$20 or diastolic BP $\geq$10 mmHg 1 minute after standing from a seated position. The association between OH and a composite of stroke, myocardial infarction, or death was assessed using an adjusted Cox regression model with OH as a time-varying variable. The association between group allocation (intervention vs. control) and OH was assessed using logistic regression. During a mean follow-up of 4.8 years, 35.3% of patients had OH at some point. OH was not significantly associated with the composite outcome (HR: 1.11, 95% CI: 0.80–1.54). Allocation to the intervention group in the NAILED trial was not associated with OH during follow-up (OR: 0.84, 95% CI: 0.62–1.13).

### Conclusions

OH was not associated with an increased risk of CV events or death in this stroke/TIA population. Systematic titration of antihypertensive treatment did not increase the prevalence of OH compared to usual care. Thus, OH did not reduce the gains of antihypertensive treatment.

**Citation:** Ögren J, Mooe T, Irewall A-L (2024) Orthostatic hypotension in stroke/TIA patients: Association with new events and the effect of the NAILED intervention. PLoS ONE 19(2): e0298435. https://doi.org/10.1371/journal.pone.0298435

**Data Availability Statement:** As open access to individual-level data was not specified in the original application approved by the ethics committee, the underlying data is only available

upon reasonable request. Data are available from the Institutional Data Access at Department of Public Health and Clinical Medicine, Umeå University (contact via registrator@umu.se) for researchers who meet the criteria for access to confidential data.

**Funding:** The study received funding from the Unit of Research, Development and Education, Region Jämtland Härjedalen, the Swedish Heart–Lung Foundation and Stroke research in northern Sweden. The funders had no role in the study design, data collection and analysis, decision to publish, or preparation of the manuscript.

**Competing interests:** The authors have declared that no competing interests exist.

## Introduction

Stroke and transient ischemic attack (TIA) patients have an increased risk of a new stroke or other cardiovascular (CV) events [1, 2]. Antihypertensive treatment decreases this risk [3] and is recommended in guidelines on secondary prevention after stroke and TIA [4, 5]. Orthostatic hypotension (OH) has been associated with an increased risk of death and CV events, including both first and recurrent stroke [6–13]. A fear of causing or worsening OH in a frail and often elderly stroke/TIA patient may lead the doctor to refrain or deaccelerate antihypertensive treatment. In randomized controlled trials (RCTs), more intensive blood pressure (BP) treatment has not been associated with an increased risk of OH [14, 15], but has not been explored in a population restricted to stroke/TIA patients.

In the Nurse-based Age-independent Intervention to Limit Evolution of Disease (NAILED) stroke trial, systematic follow-up and titration of antihypertensive medication improved the proportion of stroke and TIA patients that reached the BP target level of <140/90 mmHg [16, 17]. The intervention also decreased the risk of a new stroke [18]. The NAILED stroke trial randomized an older population with more comorbidities than commonly represented in RCTs; therefore, this population may provide increased knowledge of the incidence and prognostic importance of OH in a general stroke/TIA population.

Our aim was to assess whether OH after stroke/TIA is associated with an increased risk of new CV events or death, and whether allocation to the intervention group in the NAILED stroke trial was associated with an increased risk of OH during follow-up.

## Methods

### Study design and setting

This post-hoc analysis includes patients from the NAILED stroke trial. NAILED was a RCT that investigated whether nurse-led, telephone-based follow-up that included medication titration was more efficient than usual care in improving risk factor levels and reduce the incidence of new cardiovascular events after discharge following stroke or TIA. All patients treated for an intracerebral hematoma (ICH), ischemic stroke (IS), or TIA at Östersund Hospital between January 1, 2010, and December 31, 2013, were assessed for participation in the study. Identified patients were considered eligible if they were able to participate in telephone-based follow-up. All participants provided informed written consent of participation. In the present analysis, the NAILED stroke trial participants with registered measurements of seated and standing BP at 1 month after hospital discharge (baseline) were included. Standing blood pressure was a prespecified, secondary outcome [19] systematically collected through the course of the study to enable assessment of hypotensive as well as hypertensive reactions since both may be associated with prognosis. Related sub studies were, however, not prespecified in further detail. The present study should therefore be considered a post hoc analysis.

### Intervention

The study population was randomized (1:1) to intervention or control. All participants were followed up at 1 month after discharge and yearly thereafter. If the participants in the intervention group did not achieve the set BP target of <140/<90 mmHg, the medical treatment was adjusted. A new measurement was performed after approximately 4 weeks, and the process was repeated if necessary. Participants in the control group received secondary preventive care according to local standards. Treatment was generally initiated in-hospital. After discharge, each participant's general practitioner was responsible for the patient's care.

A more detailed description of the study design is available in the previously published study protocol [19].

## Orthostatic hypertension

OH was assessed at 1 month and then at yearly visits. Seated BP was assessed after 5 minutes in the sitting position and standing BP after 1 minute standing. OH was defined as a difference in standing and sitting systolic BP ≥20 mmHg or diastolic BP ≥10 mmHg, the same thresholds as in the consensus definition [20, 21]. Both symptomatic and asymptomatic OH was included in the analysis.

## Data collection and outcomes

Baseline data were collected during the index hospitalization and at the 1-month follow-up [19]. The participants were followed from 1 month (baseline) post-discharge until December 31, 2017. The primary outcome was the association between OH and a composite of non-fatal stroke, non-fatal type 1 acute myocardial infarction (AMI), and all-cause death. Secondary outcomes were the association between OH and a composite of non-fatal stroke, non-fatal type 1 AMI, and CV death, all stroke (ischemic or hemorrhagic excluding subarachnoid hemorrhage), IS, and all-cause death. A stroke or AMI followed by death within 30 days was considered fatal and included in the composite as CV death. The association between group allocation in NAILED and occurrence of OH during follow-up was also assessed as a secondary outcome.

All new outcome events were identified through a structured review of discharge records for hospitalizations at Östersund Hospital. Identified events were adjudicated by four experienced medical doctors. Each doctor worked separately using a standardized workflow algorithm. The process of adjudication was blinded to treatment assignment. The process and definitions of new outcome events are described elsewhere [18].

## Statistical analysis

Baseline characteristics are presented using means (standard deviation) and proportions for the overall cohort and according to the occurrence of OH during the study. To examine the association of OH and primary and secondary outcomes, a Cox proportional hazard model with OH as a time-varying covariate was used to account for the OH status changing during repeated measurements. Two adjusted models were used to account for potential confounders: Model 1 was adjusted for age and sex, and Model 2 was adjusted for age, sex, diabetes at baseline, glomerular filtration rate (GFR) <60 at baseline, number of antihypertensive agents, systolic blood pressure (SBP) at baseline, body mass index (BMI), smoking status, LDL, and treatment allocation in the NAILED stroke trial. The assumption of proportional hazards was investigated using scaled Schoenfeld residuals. Patients were followed until the occurrence of a defined outcome, death, or December 31, 2017.

To examine whether group allocation within the NAILED stroke trial was associated with occurrence of OH during follow-up, logistic regression analysis was used with the same adjusted models above. A p-value <0.05 was considered significant. SPSS version 28.0 and SAS 9.4 were used to perform statistical analyses.

## Ethics

The NAILED trial was approved by the Regional Ethical Review Board, Umeå, Sweden, on October 28, 2009 (Dnr: 09-142M), and an extended follow-up period was approved on June 10, 2013 (Dnr: 2013-204-32M).

## Results

### Baseline characteristics

The NAILED stroke trial enrolled 870 stroke and TIA patients. At the 1-month follow-up, 4 participants had died, and orthostatic BP measurements were missing for another 52 participants. Of the remaining 814 participants, 57.6% were men and the mean age was 70.7 years. The inclusion diagnosis was IS in 57.7%, TIA in 38.9%, and ICH in 3.3%. Additional baseline data for the study population overall and separately for patients with and without OH during follow-up are provided in Table 1. Orthostatic participants were older, more often diabetic, and more often had a history of hypertension and worse kidney function.

### BP at baseline and during follow-up

The mean BP in the study population at baseline was 137.2/80.1 mmHg. The mean systolic BP at each follow-up in participants with or without OH during follow-up according to allocation group (intervention or control) is shown in Fig 1. There was a trend that the mean systolic BP was higher in patients with OH compared with non-OH patients in both the intervention and control groups.

### OH during follow-up

During a median follow-up of 5.4 years (IQR: 4.3–6.4), 440 of 4098 measurements revealed OH (10.7%) and 287 of 814 patients (35%) had at least one follow-up with OH. The proportion of BP measurements showing OH was 10.3% in the intervention and 11.4% in the control group, with similar proportions at each follow-up (Fig 2).

### OH and the risk of stroke, AMI, or death

During follow-up, 276 participants had a stroke, AMI, or died, resulting in an unadjusted event rate of 7.1 (95% CI: 6.3–8.0) per 100 patient-years (Table 2). In the fully adjusted model (Model 2), there was no association between OH after stroke or TIA and the primary outcome of non-fatal stroke, non-fatal AMI, or all-cause death (HR: 1.11, 95% CI: 0.80–1.54). Furthermore, there was no association between OH and any of the secondary outcomes (Table 3).

### Association of group allocation and OH

Allocation to the NAILED intervention group was not associated with an increased risk of having OH during follow-up (OR: 0.84, 95% CI: 0.62–1.13). A fully adjusted model did not change the result.

## Discussion

In this post-hoc analysis of the NAILED stroke trial, we studied the association between OH and incidence of stroke, AMI, or death in a stroke and TIA population. We found that OH was not associated with any of these outcomes. We also studied the association between group allocation in NAILED and OH and found no indication of increased prevalence of OH when blood pressure was managed according to the NAILED intervention. During a median follow-up of 5.4 years, 35% of the study population had at least one follow-up with OH after a stroke or TIA. Patients with OH were older and risk factors for CV disease more common. These differences were adjusted for in the primary result.

The association of OH and CV events or death has been investigated previously in various populations, with various findings. In prospective population cohorts, an association between

**Table 1. Baseline characteristics according to OH during follow-up.**

| | All (n = 814) | No OH (n = 527) | OH (n = 287) |
|---|---|---|---|
| Age, years, mean (SD) | 70.7 (10.8) | 69.7 (11.2) | 72.5 (9.9) |
| Female | 345 (42.4) | 223 (42.3) | 122 (42.5) |
| Current or prior smoking | 416 (51.1) | 264 (50.1) | 152 (53.0) |
| Body mass index, kg/m$^2$, mean (SD) | 26.8 (4.5) | 26.9 (4.6) | 26.7 (4.4) |
| mRS >2 at discharge | 205 (25.2) | 125 (23.7) | 80 (27.9) |
| **Comorbidities** | | | |
| GFR <60%[1] | 208 (25.6) | 123 (23.3) | 85 (29.6) |
| Diabetes mellitus[2] | 139 (17.1) | 75 (14.3) | 64 (22.4) |
| Hypertension | 490 (60.2) | 298 (56.5) | 192 (66.9) |
| Atrial fibrillation | 145 (17.8) | 77 (14.6) | 68 (23.7) |
| Congestive heart failure | 36 (4.5) | 25 (4.7) | 11 (3.8) |
| **Medical history** | | | |
| Prior myocardial infarction | 74 (9.1) | 47 (8.9) | 27 (9.4) |
| Prior ischemic stroke | 101 (12.4) | 69 (13.1) | 32 (11.1) |
| Prior intracerebral hemorrhage | 5 (0.6) | 3 (0.6) | 2 (0.7) |
| Prior TIA | 49 (6.0) | 29 (5.5) | 20 (7.0) |
| **Index event** | | | |
| TIA | 317 (38.9) | 217 (41.2) | 100 (34.8) |
| Ischemic stroke | 470 (57.7) | 289 (54.8) | 181 (63.1) |
| Intracerebral hematoma | 27 (3.3) | 21 (4.0) | 6 (2.1) |
| **Baseline data** | | | |
| Intervention group NAILED | 407 (50.0) | 274 (52.0) | 133 (46.3) |
| No. of antihypertensive agents at baseline | | | |
| 0 | 192 (23.6) | 136 (25.8) | 56 (19.5) |
| 1 | 222 (27.3) | 141 (26.8) | 81 (28.2) |
| 2 | 214 (26.3) | 132 (25.0) | 82 (28.6) |
| ≥3 | 186 (22.9) | 118 (22.4) | 68 (23.7) |
| Statin[2] | 628 (77.3) | 409 (77.8) | 219 (76.6) |
| Antiplatelet treatment[2] | 644 (79.3) | 427 (81.2) | 217 (75.9) |
| Anticoagulant treatment[2] | 129 (15.9) | 69 (13.1) | 60 (21.0) |
| Baseline SBP, mean (SD) | 137.2 (18.5) | 135.9 (17.6) | 139.5 (19.8) |
| Baseline DBP, mean (SD) | 80.1 (11.4) | 79.8 (11.0) | 80.6 (12.2) |
| Baseline LDL,[3] mean (SD) | 2.5 (0.82) | 2.53 (0.81) | 2.44 (0.83) |
| **Number of follow-ups, mean (SD)** | 5.04 (2.06) | 4.78 (2.13) | 5.52 (1.82) |

Values are given as n (%) unless otherwise noted. mRS: modified Rankin scale; GFR: glomerular filtration rate; TIA: transient ischemic attack.

[1]Missing values for 3 participants (no OH, n = 1 and OH, n = 2).

[2]Missing values for 2 participants (no OH, n = 1 and OH, n = 1).

[3]Missing values for 4 participants, all in the no OH group.

prevalent OH and CV events, stroke, or death has been reported in several publications [6–11] but not in others [20, 21]. These publications analyze the association of OH at baseline with the risk of events during a follow-up period of several years. There is a wide range in age, degree of comorbidity, and follow-up period in these studies that could contribute to the observed differences in the results.

Some previous post-hoc analyses of RCTs have been similar to our study, in that the association between OH and CV events was explored using repeated OH measurements during

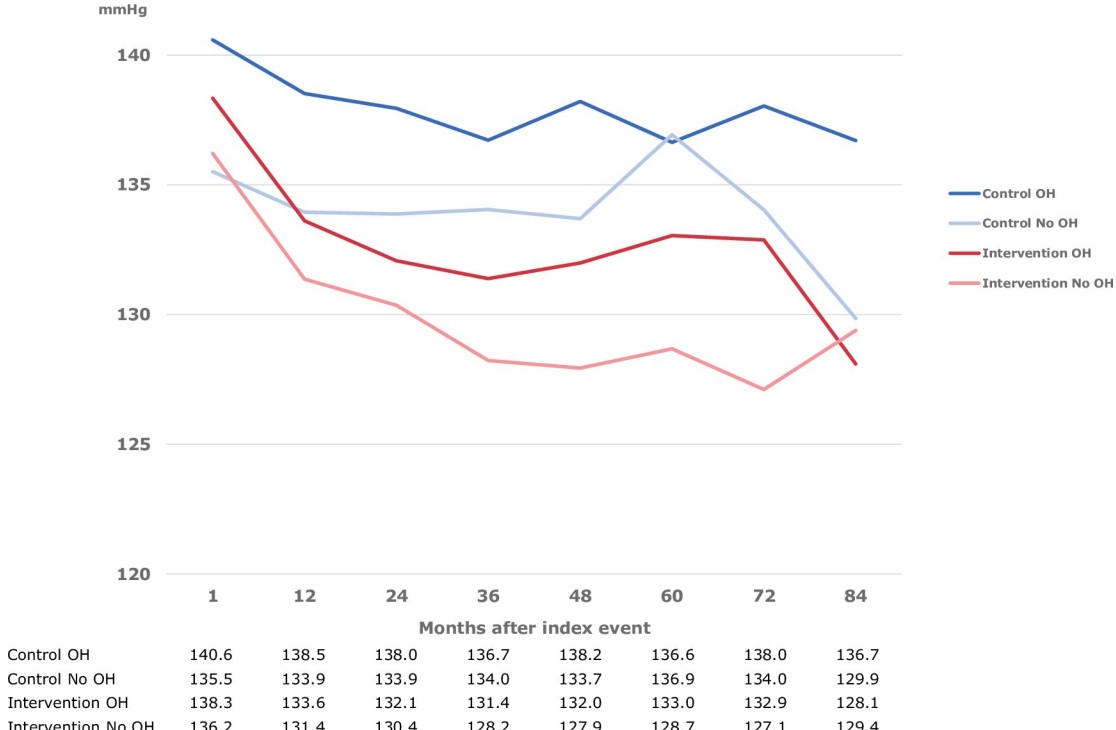

| | 1 | 12 | 24 | 36 | 48 | 60 | 72 | 84 |
|---|---|---|---|---|---|---|---|---|
| Control OH | 140.6 | 138.5 | 138.0 | 136.7 | 138.2 | 136.6 | 138.0 | 136.7 |
| Control No OH | 135.5 | 133.9 | 133.9 | 134.0 | 133.7 | 136.9 | 134.0 | 129.9 |
| Intervention OH | 138.3 | 133.6 | 132.1 | 131.4 | 132.0 | 133.0 | 132.9 | 128.1 |
| Intervention No OH | 136.2 | 131.4 | 130.4 | 128.2 | 127.9 | 128.7 | 127.1 | 129.4 |

**Fig 1. Mean systolic blood pressure (mmHg) at each follow-up by allocation group in the NAILED trial and occurrence of orthostatic hypotension (OH).**

follow-up. These trials also presented various results. In a post-hoc analysis of the systolic blood pressure trial (SPRINT), OH was not associated with increased risk of CV events among patients with high CV risk but without previous diabetes or stroke [14]. In a post-hoc analysis of the Action to Control Cardiovascular Risk in Diabetes (ACCORD) trial in patients with diabetes mellitus type 2, there was an association with OH and total death but not with the risk of AMI or stroke [22]. The SPS3 trial is the only previous study of the association between OH and new events in a stroke population. In contrast to our results, the SPS3 post-hoc analysis showed an association between OH and recurrent stroke and all-cause death. Compared to our study, the population in SPS3 was younger with less comorbidity except for a higher proportion of diabetics, and all patients had a lacunar infarct as the index event [13]. The incidence rate of new stroke in SPS3 was comparable with the present trial, whereas the incidence rate of all-cause mortality was lower, possibly because of a younger population. All stroke/TIA patients treated at Östersund Hospital were screened for participation in the NAILED stroke trial and were considered eligible if they could handle a telephone call after their hospitalization. The low threshold for participation resulted in a less selected population with more comorbidities and increased risk of new events and death compared with average RCTs.

After a postural change to standing, blood pools to the lower extremities, leading to decreased venous return and ventricular filling, resulting in reduced cardiac output and lower SBP. Baroreceptors respond to the decreased pressure, triggering an increased sympathetic tone, leading to elevated heart rate and peripheral vascular resistance [23]. If this compensatory mechanism fails, orthostatic hypotension (OH) can occur. Several mechanisms can disrupt this system, including stiff vessels as part of manifest or subclinical vascular disease [24]. OH and CV events are both associated with age, diabetes, kidney function, smoking, and

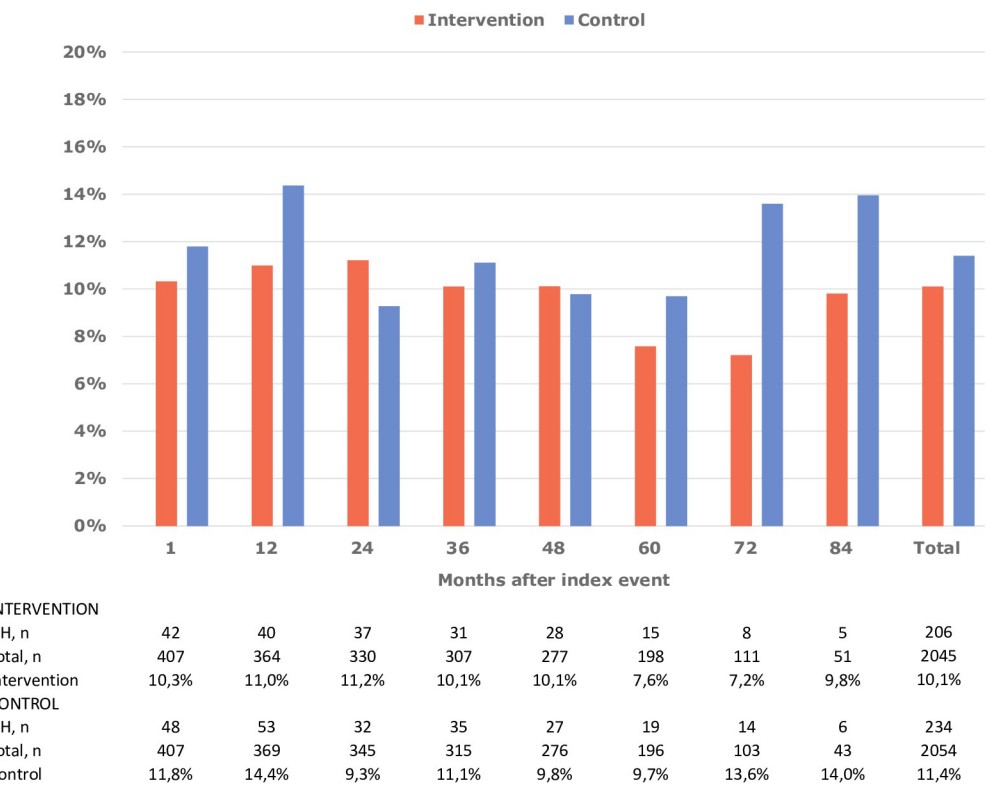

**Fig 2. Proportion of patients with orthostatic hypotension (OH) and numbers of participants at each follow-up in the intervention and control groups.**

hypertension [25–28]. In our study, the unadjusted analysis showed a trend towards an association between OH, cardiovascular events and death. However, these associations did not remain after adjusting for known risk factors of CV disease. This indicates that OH may be a marker of underlying CV disease rather than a causal risk factor of new events, and that it is important to adjust for all known risk factors of CV disease when analyzing the risk of OH.

Treating high BP after stroke or TIA decreases the risk of new CV events [3] and is important in secondary prevention. Yet, many stroke/TIA patients do not reach the treatment goal for BP [29]. Fear of OH could be a reason to deaccelerate BP treatment. Reassuringly, the present analysis showed no association of group allocation to the intervention group in the

**Table 2. Incidence of new cardiovascular events after stroke or TIA (95% confidence interval) in 814 patients according to OH during follow-up.**

| | All | | No OH | | OH | |
|---|---|---|---|---|---|---|
| | N | % / person-year | N | % / person-year | N | % / person-year |
| Stroke, AMI & all-cause death | 276 | 7.1 (6.3–8.0) | 180 | 7.4 (6.3–8.5) | 96 | 6.6 (5.3–8.0) |
| Stroke, AMI & CV-death | 182 | 4.7 (4.0–5.4) | 114 | 4.7 (3.8–5.6) | 68 | 4.7 (3.6–5.8) |
| All-cause death | 199 | 4.7 (4.1–5.4) | 135 | 5.1 (4.2–6.0) | 64 | 4.0 (3.1–5.0) |
| All stroke | 114 | 2.9 (2.4–3.4) | 72 | 2.9 (2.2–3.6) | 42 | 2.9 (2.0–3.7) |
| Ischemic stroke | 103 | 2.6 (2.1–3.1) | 66 | 2.7 (2.0–3.3) | 37 | 2.5 (1.7–3.3) |
| Myocardial infarction | 32 | 0.8 (0.5–1.0) | 19 | 0.7 (0.4–1.1) | 13 | 0.8 (0.4–1.3) |

OH: orthostatic hypotension; AMI: acute myocardial infarction; CV: cardiovascular

**Table 3. Hazard ratios (95% confidence interval) for new cardiovascular events after stroke or TIA by orthostatic hypotension.**

|  | N | Unadjusted | Model 1[1] | Model 2[2] |
|---|---|---|---|---|
| Stroke, AMI & all-cause death | 276 | 1.58 (1.14–2.19) | 1.16 (0.84–1.62) | 1.11 (0.80–1.54) |
| Stroke, AMI & CV-death | 182 | 1.55 (1.04–2.32) | 1.16 (0.77–1.74) | 1.12 (0.74–1.68) |
| All-cause death | 199 | 1.59 (1.09–2.32) | 1.14 (0.78–1.67) | 1.05 (0.72–1.55) |
| All stroke | 114 | 1.38 (0.81–2.34) | 1.09 (0.64–1.86) | 1.04 (0.61–1.78) |
| Ischemic stroke | 103 | 1.31 (0.75–2.31) | 1.05 (0.59–1.85) | 0.99 (0.56–1.75) |

TIA: transient ischemic attack; AMI: acute myocardial infarction; CV: cardiovascular

[1]Model 1: Age, sex

[2]Model 2: Age, sex, diabetes at baseline, GFR <60 at baseline, number of BP treatments, SBP at baseline, BMI, smoking status, LDL, and treatment allocation in NAILED stroke trial

NAILED stroke/TIA trial with having OH during follow-up, indicating that systematic titration of antihypertensive treatment to reach a BP <140/90 mmHg does not increase the incidence of OH. As reported previously, the intervention group had a BP 6.1 mmHg (95% CI: 3.6–8.6, p<0.001) lower at 3 years follow-up (128.1 vs. 134.2 mmHg) [16] compared to the control group and resulted in fewer new strokes [18]. Our finding, that more intense BP treatment is not associated with an increased risk of OH, is in accordance with several previous reports [15, 22, 30]. The present study did not investigate different treatment goals, but different follow-up strategies, which resulted in a significantly lower BP in the intervention group. To the best of our knowledge, this is the first study regarding the occurrence of OH after intensified BP-lowering treatment in a general population of stroke/TIA patients.

## Strengths and limitations

This trial contributes data on OH in a stroke/TIA population for which data have been sparce. A strength of this study is the relatively high mean age and prevalence of comorbidity in the study population, contributing to increased relevance to the clinical context. Another strength is the adjudication process for outcome events, leading to high-quality data being analyzed. However, there are also limitations to consider. Firstly, it is a post-hoc analysis. The NAILED trial was not dimensioned based on the outcomes analyzed in the present study. The study may therefore be underpowered, with a risk of a type b error especially in the multivariate analysis. Also, the observational design in this RCT population might be a subject to confounding. We have tried to minimize the risk by adjusting possible confounders, but we cannot exclude the possibility of residual confounding. Secondly, BP was measured at the yearly follow-ups and unable to capture possible variability between measurements. Thirdly, BP in the NAILED trial was measured in the seated position and after standing for 1 minute. The seated position could possibly underestimate the prevalence of OH. There is no uniform recommendation on the best timing of a single standing measurement, but measuring after 1 minute has not been shown to be inferior to later measurements [31, 32]. Fourthly, whether the patients with or without OH had symptoms and whether this had any impact on treatment decisions is not known. Fifthly, no analysis of current drugs to treat BP was made within this study. There is evidence that antihypertensives treatment causing sympathetic inhibition is associated with an increased risk of OH [33]. Finally, this study analyzed the association between OH and stroke, AMI, or death. Other clinically important outcomes, such as falls and fractures, were not analyzed within the framework of the present study but are important to consider in clinical practice.

## Conclusion

OH after stroke and TIA was not associated with an increased risk of stroke, AMI, or death. Furthermore, the NAILED stroke trial intervention was not associated with an increased risk of OH during follow-up compared with usual care. Thus, treating BP to a treatment goal of <140/<90 mmHg in a stroke/TIA population seems safe and fear of OH should not be a reason to deaccelerate BP treatment.

## Acknowledgments

The authors would like to thank the study nurses for their indispensable contributions to the study.

## Author Contributions

**Formal analysis:** Joachim Ögren.

**Funding acquisition:** Thomas Mooe.

**Methodology:** Joachim Ögren, Thomas Mooe, Anna-Lotta Irewall.

**Writing – original draft:** Joachim Ögren.

**Writing – review & editing:** Thomas Mooe, Anna-Lotta Irewall.

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
