## [Editor Report · Decision Letter 0]

10 Oct 2023

PONE-D-23-30870Orthostatic hypotension in stroke/TIA patients: Association with new events and the effect of the NAILED interventionPLOS ONE

Dear Dr. Ögren,

Thank you for submitting your manuscript to PLOS ONE. After careful consideration, we feel that it has merit but does not fully meet PLOS ONE’s publication criteria as it currently stands. Therefore, we invite you to submit a revised version of the manuscript that addresses the points raised during the review process.

 This is an interesting study.

There is novelty in this study. 

I give some comments from improvement.

- Please be detail with the method. Is it pre-planned ?

- Do you have data of stroke type and sub-type ? Please be detail with the radiological findings. Radiological (CT/ MRI) findings ?

- Begin the discussion with the main finding of your study. 

- Please elaborate more your findings with the biological plausibility of the result.

- Please elaborate more the limitations. Post-hoc data analyses are an issue because they conform to neither the population nor the randomization model of statistical inference. When we discover an apparent difference by way of an unplanned post-hoc analysis, we may have discovered nothing more than simple coincidence.

- The conclusion should be concise and clear. Please submit your revised manuscript by Nov 24 2023 11:59PM. If you will need more time than this to complete your revisions, please reply to this message or contact the journal office at plosone@plos.org. Please include the following items when submitting your revised manuscript:A rebuttal letter that responds to each point raised by the academic editor and reviewer(s). You should upload this letter as a separate file labeled 'Response to Reviewers'.A marked-up copy of your manuscript that highlights changes made to the original version. You should upload this as a separate file labeled 'Revised Manuscript with Track Changes'.An unmarked version of your revised paper without tracked changes. You should upload this as a separate file labeled 'Manuscript'.

We look forward to receiving your revised manuscript.

Kind regards,

Rizaldy Taslim Pinzon

Academic Editor

PLOS ONE

Journal Requirements:

2. Please amend the manuscript submission data (via Edit Submission) to include author Thomas Mooe and Anna-Lotta Irewall.

Additional Editor Comments:

This is an interesting study.

There is novelty in this study.

I give some comments from improvement.

- Please be detail with the method. Is it pre-planned ?

- Do you have data of stroke type and sub-type ? Please be detail with the radiological findings. Radiological (CT/ MRI) findings ?

- Begin the discussion with the main finding of your study.

- Please elaborate more your findings with the biological plausibility of the result.

- Please elaborate more the limitations. Post-hoc data analyses are an issue because they conform to neither the population nor the randomization model of statistical inference. When we discover an apparent difference by way of an unplanned post-hoc analysis, we may have discovered nothing more than simple coincidence.

- The conclusion should be concise and clear.

---

## [Author Response · Author response to Decision Letter 0]

1 Nov 2023

Dear Dr. Pinzon

Thank you for comments and the opportunity to resubmit an improved manuscript. Clarifications and additions to the manuscript have been made according to your useful suggestions and we hope that the manuscript now is more comprehensive. All changes is described in the document "response to reviewers" and also highlighted in "revised manuscript with tracked changes".

---

## [Decision Letter · Decision Letter 1]

27 Dec 2023

PONE-D-23-30870R1Orthostatic hypotension in stroke/TIA patients: Association with new events and the effect of the NAILED interventionPLOS ONE

Dear Dr. Ögren,

Thank you for submitting your manuscript to PLOS ONE. After careful consideration, we feel that it has merit but does not fully meet PLOS ONE’s publication criteria as it currently stands. Therefore, we invite you to submit a revised version of the manuscript that addresses the points raised during the review process.

This is  a good study. 

There is novelty in this study. 

I give some inputs based on the reviewers comments. 

- Please be more detail in the method section about orthostatic hypotension.Definition of OH should include symptoms. please add a reference for this Subgroup analysis for stroke vs. hematoma should be added It should be added how variables were chosen Performance of multivariate model should be added

- Please begin the discussion with the main findings of your study.

- Conclusion should be concise and clear.  Please submit your revised manuscript by Feb 10 2024 11:59PM. If you will need more time than this to complete your revisions, please reply to this message or contact the journal office at plosone@plos.org. Please include the following items when submitting your revised manuscript:A rebuttal letter that responds to each point raised by the academic editor and reviewer(s). You should upload this letter as a separate file labeled 'Response to Reviewers'.A marked-up copy of your manuscript that highlights changes made to the original version. You should upload this as a separate file labeled 'Revised Manuscript with Track Changes'.An unmarked version of your revised paper without tracked changes. You should upload this as a separate file labeled 'Manuscript'.

We look forward to receiving your revised manuscript.

Kind regards,

Rizaldy Taslim Pinzon

Academic Editor

PLOS ONE

Additional Editor Comments:

This is a good study.

There is novelty in this study.

I give some inputs based on the reviewers comments.

- Please be more detail in the method section about orthostatic hypotension.Definition of OH should include symptoms. please add a reference for this Subgroup analysis for stroke vs. hematoma should be added It should be added how variables were chosen Performance of multivariate model should be added

- Please begin the discussion with the main findings of your study.

- Conclusion should be concise and clear.

Reviewers' comments:

Reviewer's Responses to Questions

**Comments to the Author**

1. If the authors have adequately addressed your comments raised in a previous round of review and you feel that this manuscript is now acceptable for publication, you may indicate that here to bypass the “Comments to the Author” section, enter your conflict of interest statement in the “Confidential to Editor” section, and submit your "Accept" recommendation.

Reviewer #1: All comments have been addressed

Reviewer #2: (No Response)

2. Is the manuscript technically sound, and do the data support the conclusions?

Reviewer #1: Yes

Reviewer #2: Yes

3. Has the statistical analysis been performed appropriately and rigorously? 

Reviewer #1: Yes

Reviewer #2: Yes

4. Have the authors made all data underlying the findings in their manuscript fully available?

Reviewer #1: Yes

Reviewer #2: Yes

5. Is the manuscript presented in an intelligible fashion and written in standard English?

Reviewer #1: Yes

Reviewer #2: Yes

6. Review Comments to the Author

Reviewer #1: The paper, having undergone a previous revision, has been meticulously reviewed, with each recommendation from the previous reviewer conscientiously addressed. The overall quality of the paper is commendable, characterized by clear and concise writing. However, it is worth noting that a minor oversight occurred in the manuscript. Specifically, in one instance, the term 'Model 2' was omitted where it should have been included. The sentence in question reads, 'Model 1 was adjusted for age and sex, and Model was adjusted for age, sex, diabetes at baseline.' I suggest that, the correct wording should be 'Model 1 was adjusted for age and sex, and Model 2 was adjusted for age, sex, diabetes at baseline.' This is the only discrepancy identified.

Reviewer #2: This is an interesting post hoc analysis trying to assess the impact of OH on outcomes in a stroke population. Some issues need to be addressed

Abstract/methods: it should be added if this is a prespecified analysis or not

Methods: Definition of OH should include symptoms. please add a reference for this

Methods. Subgorup analysis for stroke vs. ematoma should be added

Methods. It should be added how variables were choosen

Methods: performance of multivariatre model should be added

Adjiducagtion of OH: it should be added when was performed and at which timeline

7. PLOS authors have the option to publish the peer review history of their article (what does this mean?). If published, this will include your full peer review and any attached files.

Reviewer #1: No

Reviewer #2: **Yes: **Fabrizio D'Ascenzo

---

## [Author Response · Author response to Decision Letter 1]

11 Jan 2024

Reviewer #1 

COMMENT: The paper, having undergone a previous revision, has been meticulously reviewed, with each recommendation from the previous reviewer conscientiously addressed. The overall quality of the paper is commendable, characterized by clear and concise writing. However, it is worth noting that a minor oversight occurred in the manuscript. Specifically, in one instance, the term 'Model 2' was omitted where it should have been included. The sentence in question reads, 'Model 1 was adjusted for age and sex, and Model was adjusted for age, sex, diabetes at baseline.' I suggest that, the correct wording should be 'Model 1 was adjusted for age and sex, and Model 2 was adjusted for age, sex, diabetes at baseline.' This is the only discrepancy identified.

RESPONSE: Thank you for noticing the missing “2”. This is now adjusted (page 5, last paragraph). 

Reviewer #2 

COMMENT: Abstract/Methods: it should be added if this is a prespecified analysis or not

RESPONSE: The collection of standing blood pressure was prespecified and considered as a secondary outcome while this post-hoc analysis was not prespecified in detail. This is explained in the methods (page 4, first paragraph).

COMMENT: Methods: Definition of OH should include symptoms. please add a reference for this

RESPONSE: Consensus documents define OH as a drop in standing systolic blood pressure of at least 20 mmHg or diastolic blood pressure of 10 mmHg. OH, can be symptomatic or asymptomatic. This study included both symptomatic and asymptomatic OH. This is clarified and references are given in the Methods section (page 4, last paragraph and page 5, first paragraph).

COMMENT: Methods. Subgorup analysis for stroke vs. ematoma should be added

RESPONSE: We are not sure that we understand exactly what subgroup analysis reviewer 2 would like to be performed. However, with 27 patients with intracerebral hematoma at baseline and 9 outcome-events of stroke that was not ischemic, there will be too few cases for any meaningful subgroup analysis.

COMMENT: Methods. It should be added how variables were choosen

RESPONSE: Baseline variables were collected during the index hospitalization and at the 1-month follow-up according to the NAILED study protocol. A reference for this is included in the manuscript and for clarity now also added after the description of baseline data (page 5, second paragraph): “[19]”. A reference for the process and definitions of new cardiovascular events is also included in the manuscript. 

COMMENT: Methods: performance of multivariatre model should be added

RESPONSE: The multivariable models are described in the statistical analysis section (page 5, last paragraph) and the results are presented in table 3.

COMMENT: Adjiducagtion of OH: it should be added when was performed and at which timeline

RESPONSE: Measurement of seated and standing BP was performed 1 month after discharge of index event and then at the yearly visits in the NAILED trial. This is explained in the manuscript (page 4, last paragraph).

---

## [Editor Report · Decision Letter 2]

24 Jan 2024

Orthostatic hypotension in stroke/TIA patients: Association with new events and the effect of the NAILED intervention

PONE-D-23-30870R2

Dear Dr. Orgen

We’re pleased to inform you that your manuscript has been judged scientifically suitable for publication and will be formally accepted for publication once it meets all outstanding technical requirements.

Kind regards,

Rizaldy Taslim Pinzon

Academic Editor

PLOS ONE
---

## [Editor Report · Acceptance letter]

14 Feb 2024

PONE-D-23-30870R2 

PLOS ONE

Dear Dr. Ögren, 

I'm pleased to inform you that your manuscript has been deemed suitable for publication in PLOS ONE. Congratulations! Your manuscript is now being handed over to our production team.

Kind regards, 

on behalf of

Dr. Rizaldy Taslim Pinzon 

Academic Editor

PLOS ONE